# Sodium Alginate-g-acrylamide/acrylic Acid Hydrogels Obtained by Electron Beam Irradiation for Soil Conditioning

**DOI:** 10.3390/ijms24010104

**Published:** 2022-12-21

**Authors:** Elena Manaila, Gabriela Craciun, Ion Cosmin Calina

**Affiliations:** Electron Accelerators Laboratory, National Institute for Laser, Plasma and Radiation Physics, 409 Atomistilor St., 077125 Magurele, Romania

**Keywords:** sodium alginate, hydrogels, swelling, water loss studies, electron beam irradiation

## Abstract

Being both a cause and a victim of water scarcity and nutrient deficiency, agriculture as a sustainable livelihood is dependent now on finding new suport solutions. Biodegradable hydrogels usage as soil conditioners may be one of the most effective solutions for irrigation efficiency improvement, reducing the quantity of soluble fertilizers per crop cycle and combating pathogens, due to their versatility assured by both obtaining method and properties. The first goal of the work was the obtaining by electron beam irradiation and characterization of some Sodium Alginate-g-acrylamide/acrylic Acid hydrogels, the second one being the investigation of their potential use as a soil conditioner by successive experiments of absorption and release of two different aqueous nutrient solutions. Alginate-g-acrylamide/acrylic Acid hydrogels were obtained by electron beam irradiation using the linear accelerator ALID 7 at 5.5 MeV at the irradiation doses of 5 and 6 kGy. For this were prepared monomeric solutions that contained 1 and 2% sodium alginate, acrylamide and acrylic acid in ratios of 1:1 and 1.5:1, respectively, for the obtaining of materials with hybrid properties derived from natural and synthetic components. Physical, chemical, structural and morphological characteristics of the obtained hydrogels were investigated by specific analysis using swelling, diffusion and network studies and Fourier Transform Infrared Spectroscopy and Scanning Electron Microscopy. Four successive absorption and release experiments of some synthetic and natural aqueous solutions with nutrients were performed.

## 1. Introduction

While the human need for water is approximately 2 L per day, the water required for the production of daily food for each person is 3000 L [1,2]. About 70% of the planet’s fresh water reserves are used only in agriculture for irrigation, the application of pesticides and fertilizers or animal husbandry [1,3]. The population growth in recent decades has led to an increase in the demand for drinking water and food. From this perspective, agricultural production will be expanded by 70% by 2050, a fact that will require a careful management of water resources [1]. Additionally, the obvious climatic changes manifested through the lack of rainfall have forced a return, where was possible, to agriculture based on irrigation and to the development of other systems to replace the old rain-fed crop system. This was one of the reasons of the development of some superabsorbent materials in order to ensure the efficient use of water, and the maintenance and durability of the soil [3].

Hydrogels are three-dimensional cross-linked polymers capable of absorbing a large amount of water up to a hundred times their dry weight, without dissolving [4]. The use of hydrogels in agriculture represents an approach to reducing water consumption in conditions of water stress to maintain soil moisture in the rooting zone of crops by reducing evaporation, deep percolation and runoff losses. They can absorb a huge amount of water in conditions of abundant precipitation or during irrigation that can be gradually released when the soil area around the roots of the plants dries out in drought conditions. In additional to the water stress reduction, they may have the ability to improve the physico-chemical, hydrophysical and biological properties of the soil, increase the efficiency of the use of water and nutrients and increase the yield and quality of crops [5].

Based on the source materials, hydrogels for agricultural use are classified into three types: natural, synthetic and semi-synthetic [5,6]. Synthetic hydrogels are based on monomers derived from petroleum that are attractive for the synthesis of polymeric hydrogels, due to their very controllable physical and chemical properties compared to natural polymers [7,8]. The most used synthetic polymers in the synthesis of hydrogels are poly (vinyl alcohol) (*PVA*), poly (ethylene glycol) (*PEG*), poly (ethylene oxide) (*PEO*), poly (2-hydroxyethyl methacrylate) (*PHEMA*), poly (acrylic acid) (*PAA*) and poly(acrylamide) (*PAAm*) [7]. Synthetic polymer hydrogels have high water retention capacity, longer shelf life and improved gel strength [9,10]. Natural biopolymers that form hydrogels, such as cyclodextrin, dextran, chitosan, agarose, sodium alginate, fibrin, hyaluronic acid, gelatin and collagen, are derived from natural sources, so are abundant, non-toxic, cheap and biodegradable [8,9]. On the other hand, they show poor mechanical strength [7].

Another way of classifying hydrogels, regardless of the origin source, is based on the obtaining method; the cross-linking can be physical (by temporary non-covalent bonds), chemical (by permanent covalent bonds) or combined [7]. By physical cross-linking through hydrogen bonding, ionic forces, Van der Waals interactions, stereocomplexation and hydrophobic forces [7], hydrogels with reversible properties, a short life span and with poor mechanical properties are obtained [8]. For these reasons, physical cross-linked hydrogels are not proper for agricultural use. Instead, chemical cross-linking leads to the formation of permanent hydrogels, with mechanical and thermal stability and controllable properties, that are ideal for reducing water and nutritional stress in soils [8]. The obtaining of chemical hydrogels, not so dependent on pH as in the case of physical hydrogels, involves chemical reactions, polymerization, homopolymerization and grafting [7,8]. The classic chemical cross-linking methods involve the mandatory completion of some stages, among which we mention the preparation and purification in several stages of mono- and bi-functional molecules such as formaldehyde, glutaraldehyde, genipin, diethyl squarate or ethylene glycol diglycidyl ether, and the obtained hydrogels can become cytotoxic after functionalization with reactive groups [7]. Chemical cross-linking can also be done using ionizing radiation (gamma and electron beam). The ionizing radiation technique has significant advantages, such as the ease and speed of preparation, but also the low cost of production compared to classical chemical cross-linking methods [7]. Chemically cross-linked hydrogels can also be obtained by free radical polymerization of low molecular weight monomers in the presence of cross-linking agents, the cross-linking reactions being initiated by initiators such as potassium persulfate (*KPS*), ammonium persulfate (*APS*), ceric ammonium nitrate, ferrous ammonium sulfate, 2-20-azobisisobutyronitrile (*AIBN*) or benzoyl peroxide [7].

Hybrid or semi-artificial hydrogels are obtained by the grafting of synthetic monomers on biopolymer chains. In this way, the properties of biopolymers (biodegradable, low cost and abundant sources) are added to those of synthetic polymers (mechanical resistance and controllable physico-chemical properties). Polysaccharides (cellulose, starch, chitin, chitosan, carrageenan, agarose, alginate, etc.) are examples of natural polymers that can form hydrogels. However, the vast majority of superabsorbents for agricultural applications available today are based on acrylamide and acrylic acid [4,7].

Alginates, polysaccharides extracted from brown seaweed, are linear co-polymers of D-mannuronic and L-guluronic acid monomers. In order to make the alginic acid soluble in water, different salts are incorporated, the most known and widely used compound of alginic acid being the sodium alginate that is soluble in both hot and cold water [11,12]. Sodium alginate has the ability of gel formation when is put in contact with divalent cations (Ca^2+^, Ba^2+^, Sr^2+^, etc.) [13,14]. In the presence of cations, hydrogels with a not very high swelling capacity (300–1000%) and poor mechanical properties are obtained [13,15]. Stable covalent cross-links may be introduced into alginate hydrogels using bifunctional cross-linkers, allowing for greater control over the mechanical and swelling properties of the gels [13]. To be used in agriculture, hydrogels must have certain characteristics, such as high water absorption capability, a rate of absorption and desorption capacity according to plant requirements, low soluble content and residual monomer, high durability and stability during swelling and storage, biodegradability or rewetting capability for a long time [5]. While hydrogels based on synthetic monomers (acrylic acid, polyacrylamide) are difficult to degrade in soil, hydrogels based on biopolymers are biodegradable but weak from a mechanical point of view, so they have a short lifespan, requiring frequent replacement [8].

An important need for a better plant growth is the soil quality as a major source of nutrients. Nutrient deficiency in poor or depleted soils may be due to the geographical conditions, environmental disasters or simply to the cultivation of the same agricultural crop several seasons in row. The three main nutrients needed for plant growth are nitrogen, phosphorus and potassium, but traces of oligoelements as calcium, magnesium, sulfur, iron, manganese, zinc, copper, boron and molybdenum are also needed [5,16,17,18]. For these soil conditions, hydrogels as source of water and nutrients represent a viable solution. Because the combining of the best properties of natural and synthetic polymers in order to obtain a material with hybrid properties for agriculture is topical [5,8], the first goal of the paper was the obtaining by electron beam irradiation and characterization of some Sodium Alginate-g-acrylamide/acrylic Acid hydrogels. The grafting of the synthetic monomers acrylamide and acrylic acid on the chain of the natural sodium alginate polymer was carried out with the aim of increasing the mechanical resistance of the obtained hydrogel, which allows its multiple reuse and at the same time gives it a biodegradability potential. The investigation of hydrogel’s potential use as soil conditioners by successive experiments of absorption and release of aqueous nutrient solutions was the second goal.

## 2. Results and Discussion

### 2.1. Percentages of Gel Fraction, Swelling and Equilibrium Water Content

The influence of sodium alginate (*Na-Alg*) content, ratio between the monomers acrylic acid (*AA*) and acrylamide (*AMD*) and irradiation dose on gel fraction, swelling and equilibrium water content percentages were investigated and the results are presented in Table 1.

The gel fraction (the degree of cross-linking formed in the structure of a hydrogel) of polymeric structures obtained by electron beam irradiation are usually dependent on the irradiation dose [19,20,21].

As it can be seen from Table 1, the increasing of *Na-Alg* concentration leads to the decrease of the fraction regardless of the ratio between the two synthetic monomers or the irradiation dose. *Na-Alg* is a salt form of alginic acid. Being a natural polysaccharide, by exposure to radiation it undergoes a degradation process that leads to the breaking of the main chains [22,23]. By the increasing of the irradiation dose, the average molecular weight of *Na-Alg* decreases. Potassium persulfate, used as reaction initiator of the polymerization process, is also an oxidizing agent that accelerates the degradation process of *Na-Alg*. Thus, by combining the effects of ionizing radiation and oxidizing agent, the bio-polymer splitting rate is accelerated and the dose at which the degradation occurs is reduced [24,25]. Thus, it is possible to explain the decrease of the gel fraction with the increase of *Na-Alg* concentration, at the same monomer ratio and irradiation dose. For example, at the irradiation dose of 5 kGy, H1R1-D1 presents a gel fraction of 81.39% and H2R1-D1, 79.55%. The same decrease, this time from 92.21% to 87.35%, was also recorded at the irradiation dose of 6 kGy for samples H1R2-D2 and H2R2-D2.

The decrease in the gel fraction was followed by the decrease in the percentage of water absorption. With one exception, the H2R2 mixture, the gel fraction of hydrogels with the same concentration of *Na-Alg* and the same monomer ratio, has increased with the irradiation dose. In this case, the increasing was followed by an increased of swelling. The H2R1 mixture, irradiated with 5 kGy has showed a degree of water absorption of 15,460% and irradiated with 6 kGy of 52,882%, which represents an increase of 3.42%, while the gel fraction did not show a spectacular increase, its values being 79.55% and 81.75% respectively. The spectacular increase in the degree of absorption is mainly due to the two synthetic monomers in the system, acrylamide and acrylic acid. When the mixture containing *Na-Alg*, acrylic acid and acrylamide is irradiated, an interpenetrating polymer network is formed with the chemical cross-linking of poly (*AMD*-co-*AA*) and physical cross-linking of *Na-Alg* [22,26].

From Table 1 also, it can be seen that the percentages of equilibrium water content (*EWC*) were over 99.5%, except in the same samples of the H2R1-D1 type. The double content of *Na-Alg* and the higher irradiation dose were responsible for the obtaining of a percentage of 99.81% for *EWC* (H2R1-D2).

From Figure 1a,b the influence of *Na-Alg* content on hydrogels swelling percentages can be seen. Hydrogels containing 1 g of *Na-Alg* (1%) did not reach swelling over 29,000% (Figure 1a) and differences in swelling due to the irradiation dose were of 4500% (R1) and 2300% (R2), respectively, while hydrogels containing 2 g of *Na-Alg* (2%) reached swelling of 52,882%, and differences in swelling due to the irradiation dose were of 37,400% (R1) and 14,100% (R2) respectively.

### 2.2. Cross-Link Density, Porosity and Mesh Size

The network studies were done in order to determine the influence of sodium alginate (*Na-Alg*) content, ratio between the monomers acrylic acid (*AA*) and acrylamide (*AMD*) and irradiation dose on hydrogels cross-link density, porosity and mesh size. These parameters are determined from the equilibrium swelling experiments and are the major in defining the structure of the cross-linked hydrogel network [19,27]. The results are presented in Table 2.

As can be seen from Table 2, the increasing of the irradiation dose has led to the decreasing in cross-link density of the hydrogels even if the gel fraction has increased (Table 1). It is known that during the irradiation process, polymerization, cross-linking, grafting and degradation processes take place simultaneously. The decrease of the degree of cross-linking with the irradiation dose can be due to the degradation of the biopolymer *Na-Alg* through which the main chain splits [22,23]. Although the degree of cross-linking decreased in the case of all mixtures (H1R1, H1R2, H2R1 and H2R2) with the increase of the irradiation dose, the mesh size behaved opposite: it increased with the irradiation dose in the case of all the mixtures, a fact that shows us that the polymerization, grafting and cross-linking reactions took place in a greater proportion than the degradation ones. Lower values of mesh size after electron beam irradiation indicate a shorter distance between cross-linking points [28]. Higher mesh size values also led to an increase of swelling (Table 1). If we analyze the results from the perspective of *Na-Alg* concentration and irradiation dose, we can say that for any *Na-Alg* concentration (1% or 2%) at the same irradiation dose the cross-link density and mesh size were strongly influenced by the ratio between the two synthetic monomers, and implicitly, their concentration in the mixture. In the case of R2, which has equal amounts of the two synthetic monomers and implicitly a higher concentration of them in the mixture, the degree of cross-linking has increased, as was expected. The cross-linking and grafting reaction of the two synthetic monomers on the *Na-Alg* chain is improved due to the accessibility to several bonding possibilities. In this way, a firmer and more robust hydrogel network is formed due to the higher polymer concentration [29,30].

### 2.3. Swelling Kinetics, Swelling Power and Diffusion Coefficients Determination

The swelling of hydrogels occurs due to the diffusion of water during immersion. Diffusion involves the migration of water in formed or preexisting spaces between the hydrogel chains [31]. According to the relative rates of diffusion (*R*_diff_) and relaxation (*R*_relax_), there are three classes of diffusion [19,32,33,34]: Case I: *n* = 0.45–0.5 indicates a Fickian diffusion mechanism, in which the rate of diffusion is much smaller than the rate of relaxation (*R*_diff_ << *R*_relax_) and the system is controlled by diffusion; Case II: *n* = 1.0, where the diffusion process is much faster than the relaxation process (*R*_diff_ >> *R*_relax_) and the system is controlled by relaxation; Case III: 0.5 < *n* < 1.0 indicates a non-Fickian (anomalous) diffusion mechanism, which describes those cases where the diffusion and relaxation rates are comparable (*R*_diff_ ≈ *R*_relax_). Occasionally, when *n* > 1, the situation is regarded as Super Case II kinetics [19,32,35,36]. The results of the tested kinetic model, ln*W* vs. *t*, are presented in Figure 2a,b.

As can be seen from Figure 2, the first order swelling equation seems to fit very well. The increasing of the irradiation dose does not lead to the same improvement of swelling rate of hydrogels of H1R1 and H1R2 types corresponding to 1% *Na-Alg* (Figure 2a) as in the case of the hydrogels of H2R1 and H2R2 types corresponding to 2% *Na-Alg* (Figure 2b). There can be seen a significant difference in swelling rate of hydrogels obtained at the irradiation dose of 5 kGy (D1) independent of the ratio between *AMD* and *AA* (R1 or R2) and the one of hydrogels obtained at the irradiation dose of 6 kGy (D2) for which the ratio between *AMD* and *AA* influences the result in favor of a greater content of *AMD* (Figure 2b).

In order to calculate the swelling rate constant (*k*_1,S_/min^−1^), diffusion exponent (*n*), diffusion constant (*k*) and diffusional coefficient (*D*), the ln*F* vs. ln*t* and *F* vs. *t*^1/2^ are plotted and the results are presented in Figure 3 and Figure 4 and Table 3.

It is known that the high water content of hydrogels is represented by the bulk water, which is similar to the bulk water coming from outside of the gel. In a polymeric network of hydrogels, there are at least three types of water structures, as follows: bulk water, primary and secondary bound-water. When a hydrogel is immersed in water, the first type of water that will be present is the primary bound-water, due to the hydration of hydrophilic groups of the polymer. The primary bound-water is very difficult to remove from the gel. As a result, the network swells and the hydrophobic groups are exposed, leading to the formation of secondary bound-water [37,38,39].

As can be seen from Table 3, the values of the diffusion exponent (*n*) vary between 0.881 and 0.968, which corresponds to a diffusion mechanism of non-Fickian type, in which the diffusion and relaxation rates are comparable (*R*_diff_ ≈ *R*_relax_). Only for the H2R1-D2 samples an over unit value of n (1.093) corresponding to a transport mechanism of Super Case II or Case II (controlled relaxation) type was obtained. The results are similar to those obtained in other studies (non-Fickian characteristic): 0.353 < *n* > 1.136 for hydrogels based on chitosan and sodium alginate [32]; 0.58 < *n* > 0.69 for poly(acrylamide-co-crotonic acid) hydrogels [40]; 0.597 < *n* > 0. 795 in distilled water for poly(acrylamide-co-acrylic acid) hydrogels [27].

The calculated diffusion coefficients of water (*D*) are between 10^−1^–10^−2^ cm^2^ min^−1^. In the case of cylindrical hydrogels, the diffusion coefficients are normally obtained in the order of 10^−7^, but in the present case, relatively higher diffusion coefficients indicate a faster penetration of water molecules into the hydrogel network. The relatively large size of hydrogel meshes (Table 2) may also contribute to the faster penetration of water [41]. On the other hand, the faster transport of water in the hydrogel network can also be caused by the osmotic pressure of the gel that must be permanently overcome by water during the swelling process. Thus, the osmotic pressure may be responsible for the deviation of the water transport mechanism from Fickian to non-Fickian behavior [42,43].

The highest diffusion coefficients (*D*) were obtained for for H1R1 independent of the irradiation dose. One more time, the *Na-Alg* content influences the result (5.070 for H1R1-D2 and 4.321 for H1R1-D2, respectively).

### 2.4. Absorption and Release Capacities

In order to use the hydrogels as soil conditioners or as a system for water and nutrients delivery in poor soils they should meet, among other conditions, the following: high absorption capacity, durability and stability during swelling and to be usable over as many absorption–release cycles as possible [5]. With this purpose, successive absorption–release experiments were done at 25 °C using two different aqueous solutions with nutrients. Differences between the two solutions taken in the experiments were in terms of nature and pH: Solution no. 1, the exclusively synthetic one, was acidic with a pH of 5.4, while the Solution no. 2, the natural one, was basic with a pH of 7.45. Hydrogels were immersed in the aqueous solutions with nutrients until the constant mass was reached. Then were extracted and left to dry at a room temperature of 25 °C until the constant mass had been reached again. After drying, hydrogels were re-immersed in the aqueous solutions with nutrients, the previously described procedure being repeated four times.

The swelling and mass loss in the solutions were evaluated in each of the four swelling–release cycles, and the results are presented in Figure 5 and Figure 6. The duration of swelling in every cycle was 48 h and the release was about 216 h.

As can be seen from Figure 5a, with each swelling cycle in Solution no. 1, the hydrogels have swelled more, the absorption maximum being reached in cycles 3 and/or 4. The biggest absorptions were registered in the case of H1R1-D2 (10,271%), H2R1-D2 (9412%) and H2R2-D1 (9136%), these being the hydrogels with the biggest mesh sizes: 611, 688 and 844 nm, respectively. Independent of the irradiation dose and ratio between *AMD* and *AA*, the swelling behavior of hydrogels containing 2% *Na-Alg* is a constant one in each of the four swelling cycles. From Figure 6a, it can be seen that the mass loss was increasingly bigger with each swelling cycle reaching a maximum of 25.26% for the same sample H1R1-D2. It looks as though the acidic pH of the Solution no.1 has led to this result.

In Solution no. 2 (Figure 5b), the hydrogels have showed a behavior near the expected one: after the fourth swelling–drying cycle, the swelling at equilibrium has decreased. The maximum absorption was reached in cycle 1 or 2, with the exception of two hydrogels whose swelling continues to increase until cycle 3 (H2R2-D2) and cycle 4 (H1R2-D1), these being the ones with the smallest mesh sizes: 431 and 371 nm, respectively. The maximum swelling at equilibrium was of 26,855% (H2R2-D2) and 29,386% (H2R1-D2) for hydrogels irradiated at the irradiation dose of 6 kGy, containing 2% *Na-Alg* and with different quantities of *AMD* and *AA* and mesh sizes of 431 and 688 nm, respectively. The mass loss behavior in Solution no. 2 was completely different from the case of Solution no. 1 as it can be seen from Figure 6b. Some components from the natural solutions were retained in the hydrogels, except in the cases of two hydrogels irradiated at the irradiation dose of 6 hydrogels, H1R1-D2 and H2R1-D2. Evaluation of absorption and release capacity by multiple usage experiments was made using the hydrogels that have presented the smallest and the biggest gel fraction percent, H2R1-D1 (79.55%) and H1R2-D2 (92.21%), respectively. The experiments were carried out in both types of solutions, synthetic and natural, and the results are presented in Figure 7. It can be observed that all the hydrogels tested show values of absorption degree over 5000% in both solutions. These gels swelled up more due to the presence of *AMD*, *AA* and *Na-Alg*, which contain amine groups (that results in more hydrophilic groups, such as OCOO and OCONH_2_), hydroxyl (-OH) and COOH groups that are also hydrophilic [44]. The lowest degree of swelling obtained in Solution no. 1 (pH = 5.4) may be due to the protonation of the -NH- bond of the amide in *AMD* and, therefore, to the increase in the resistance of association with water through H bonds. In Solution no. 2 (pH = 7.45) deprotonation takes place, and as a result the degree of swelling increases [45].

As it can be seen from Figure 7a,b, both hydrogels have presented similar behaviors, except in H1R2-D2 in the fourth absorption and release cycle in Solution no. 2.

In Figure 8 and Figure 9 are presented photographs of the two hydrogels previously taken in discussion, H2R1-D1, having the gel fraction of 79.55%, and H1R2-D2, having the gel fraction of 92.21%, during the four absorption–release experiments.

### 2.5. Spectral Characterization

The structural changes that appeared after electron beam irradiation in dried hydrogels before and after the swelling process were investigated by ATR-FTIR. The position of functional groups and bands assignments for sodium alginate, acrylamide and hydrogels are collected in Table 4. Being in liquid form, *AA* was not investigated as was done for *Na-Alg* and *AMD*. However, specific known bands were identified in the hydrogels, meaning that the grafting of *AA* on *Na-Alg* had been produced.

The characteristic FTIR spectra of hydrogels obtained by irradiation at doses of 5 and 6 kGy are showed in Figure 10. It can be seen that the intensity of the absorption bands has varied with the irradiation dose and the concentration of *Na-Alg* and *AA*/*AMD* ratio. It is obvious that the absorption band around 3330–3340 cm^−1^, assigned to the stretching vibration of the NH group in the *AMD*, shows a slight broadening and shift to lower wavenumbers when the content of *Na-Alg* increases.

The position of bands assigned to the O-H groups was shifted from 3190 cm^−1^ to 3200 cm^−1^ for H1R1-D1 hydrogel and 3202 cm^−1^ for H1R2-D1 hydrogel after 5 kGy, while the bands for the H2R1-D1 and H2R1-D2 hydrogels were lower (3198 cm^−1^; 3196 cm^−1^) [46,47,48,49,50]. The bands from 2935–2938 cm^−1^ are related to the stretching vibrations of CH_2_. As the ratio of *Na-Alg* and *AMD*/*AA* ratio increases (R1), the intensity of these absorption bands (3445–2850 cm^−1^ range) decreases, except for the intensity of the H2R1-D2 hydrogel obtained by irradiation at 6 kGy. Therefore, new bonds are formed between the carboxyl groups of *Na-Alg* and the -NH groups of *AMD* [50]. The two bands corresponding to the C=O stretching vibration, COO^-^ symmetric bending and N-H bending vibration shifted to appear at 1654 cm^−1^ and 1607 cm^−1^ [50]. The band at 1448 cm^−1^ (COO^-^) remained the same for all hydrogel compositions. The band located at 1320 cm^−1^ is usually attributed to the N-H stretching vibration that was shifted to 1318 cm^−1^ after irradiation [46,47,48,49,50]. Other significant changes of the hydrogels were observed for the bands located around 1120 cm^−1^ (C-O-C), whose intensities decreased with the irradiation dose and *AMD*/*AA* ratio (R1) [50]. The band at 1030 cm^−1^ was drastically shifted to a higher wavenumber at 1043 cm^−1^ for the H2R2-D2 hydrogel. The increasing of band intensities and their shifting towards lower wavenumbers demonstrate this can be associated with the cross-linking of the hydrogel network. These results are correlated with the results obtained for gel fraction and swelling degree.

Figure 11 shows the FTIR spectra of hydrogels after the swelling process in the aqueous solutions with nutrients.

Significant changes in wavenumber, intensity and morphology in the bands of the functional groups were found in hydrogels that had been swollen: -NH (3336/3304 cm^−1^); -OH (3195–3290 cm^−1^); CH_2_ (2924–2938/2854–2855 cm^−1^); C=O (1644–1659 cm^−1^); COO^−^ (1542–1556/1438–1454 cm^−1^); N-H (1314–1353 cm^−1^) and C-O (1037–1059 cm^−1^) [46,47,48,49,50]. These modifications suggest changes in the chemical structures of hydrogels, such as ion exchange or the formation of new bonds that justify the increase of the multiple swelling performances of the hydrogels [46,47,48,49,50].

The formation of radicals by decomposition of all reactants constitutes an advantage of the radiation-induced reactions. Decomposition of the reactants is presented in Figure 12: solvent (water radiolysis) (Figure 12a), acrylamide (Figure 12b), acrylic acid (Figure 12c) and sodium alginate (Figure 12d) [51,52].

In order to correlate the results obtained by FTIR analysis, we propose a possible reaction mechanism that highlights how the *AMD* and *AA* is bound to the *Na-Alg* structure. The reaction mechanisms for *AMD* and *AA* grafting on *Na-Alg* (Figure 13) are based on the obtained results and on literature studies [47,49,53].

The proposed mechanism is argued by the results presented in Table 4 and in the FTIR spectra (Figure 10) in which bands, due to the asymmetric and symmetric stretching vibrations of -NH and -CN from *AMD* and of C-H and CH_2_ exclusive from *Na-Alg*, were found more or less shifted in the hydrogels spectra, a fact that confirms the grafting [47,49,53].

### 2.6. Morphological Characterization

SEM investigations were done on the same hydrogels used in the absorption–release experiments: H2R1-D1 having the smallest gel fraction (79.55%) and H1R2-D2 having the biggest gel fraction (92.21%), respectively. As can be seen from Figure 14, SEM images present dense and smooth surfaces.

In Figure 15 and Figure 16 are presented SEM micrographs of H2R1-D1 and H1R2-D2 hydrogels dried after the fourth absorption–release cycle in both types of aqueous solution with nutrients.

Surfaces presented in Figure 15a1,b1 and Figure 16a1,b1 are rugged and loose with uneven interspaces, favorable for water molecules to get inside [54,55], an aspect that correlates with the high degrees of absorption in both types of solutions. For a better view of the rough appearance of the surfaces, we have chosen to present images captured at the magnifications of 1000 (a2 and b2) and 5000 (a3 and b3). The increased sizes of crack-like structures that provide more space for solution penetration correlated with the high degree of absorption that can be seen [54,55].

## 3. Materials and Methods

### 3.1. Materials

In Table 5 are listed the materials, purchased from Redox Group Company, Bucharest, Romania, Sigma-Aldrich, Saint Louis, Missouri, US representative, excepting TMPT that was purchased from Lehmann&Voss&Co Hamburg, Germany.

In Table 6 are listed the aqueous nutrient solutions used for hydrogels testing in terms of their multiple use.

### 3.2. Experimental Installation and Samples Preparation

Hydrogels were obtained by electron beam irradiation using the installation of national interest (IOSIN) called ALID 7, which is a linear accelerator of 5.5 MeV built in the Electron Accelerators Laboratory from the National Institute for Lasers, Plasma and Radiation Physics, Bucharest, Romania. The nominal values of the electron beam (EB) parameters were as follows: energy of 5.5 MeV, peak current of 130 mA, output power of 134 W, pulse duration of 3.75 μs and pulse repetition frequency of 50 Hz [19,56,57]. The irradiation process performance depends on the rigorous control of the irradiation dose and dose rate. In our experiments, the process dose rate was of 3 kGy/min. The primary standard graphite calorimeter was used for radiation dosimetry. In order to ensure equality between the doses at the entrance and exit of the irradiated sample and for an efficient use of the electron beam, the penetration depth was calculated according with the following equation [19,56,57].
(1)E=2.6⋅t⋅ρ+0.3
where *E* (MeV) is the electron beam energy, *t* (cm) is the sample thickness, and *ρ* (g·cm^−3^) is the sample density (in our case, 1 g·cm^−3^). The proper thickness of samples subjected to EB irradiation was calculated as being 20 mm [19,56,57].

For irradiation, four types of aqueous solutions were prepared (Table 7). The monomeric solutions were places in medical syringes with a diameter of 1.5 mm and irradiated using electron beam doses of 5 and 6 kGy in atmospheric conditions and at a room temperature of 25 °C.

The meaning of the codes in the Table 7 is as follows: H1 and H2 were assigned to the *Na-Alg* content (1 and 2 g of *Na-Alg* per 100 mL of solution, respectively); R1 and R2 represent the ratios between the monomers acrylamide (*AMD*) and acrylic acid (*AA*) that were of 1.5 (R1) and 1(R2), respectively; D1 and D2 are attributed to the two irradiation doses of 5 and 6 kGy, respectively.

### 3.3. Sodium Alginate-g-acrylamide/acrylic Acid Hydrogels Characterization

Before making any measurements, the hydrogels were visually evaluated as being hardened enough by the electron beam treatment. All the samples used in the swelling and uptake tests have remained intact, and did not break or shatter.

After irradiation, the obtained hydrogels were left at room temperature for 24 h and then were cut into round pieces with thicknesses of 3–4 mm, dried in the air for three days and then in a laboratory oven at 50 °C for 24 h to reach a constant weight. They were then stored in desiccators. Dried hydrogels were used for spectral characterization and for determining of swelling, diffusion and network parameters.

For swelling and diffusion experiments, dried and weighed hydrogels were immersed in double-distilled water at a room temperature of 25 °C in a water bath. The amount of absorbed water was determined by weighing at precise time intervals, until they reached a constant mass, using the electronic balance of HR 200 type with a precision of 0.1 mg.

#### 3.3.1. Gel Fraction, Swelling and Equilibrium Water Content Measurements

First, samples prepared as before were immersed in distilled water for 24 h at room temperature for removing the unreacted soluble fraction. Then the samples were taken out, dried in air and finally dried in a vacuum oven for 24 h at 50 °C to constant weight. The gel fraction was calculated as follows [58]:(2)$$Gel-fraction(\%)=\frac{m_{f}}{m_{i}}\times 100$$
where *m_i_* is the initial weight of the dried sample and *m_f_* is the weight of sample after extraction from the water and drying [19].

In order to determine the maximum water uptake of Sodium Alginate-g-acrylamide/acrylic Acid hydrogels, swelling experiments have been done in double-distilled water at a room temperature of 25 °C. The mass increasing was evaluated by regular weighing until the equilibrium had been reached.

A fundamental relationship exists between the swelling of the polymer in a given solvent and between its nature and solvent. The swelling *S* (%) was calculated from the following relation [20,31,59]:(3)$$S(\%)=\frac{M_{t}-M_{0}}{M_{0}}\times 100$$
where *M_t_* is the mass of the swollen gel at time *t* and *M_0_* is the initial mass of the dried gel (at time *t* = 0) [19].

Another parameter used for the assessment of hydrogels swelling is the percentage of equilibrium water content (*EWC*%) that can be calculated from following equation [31,60]:(4)$$EWC=\frac{M_{S}-M_{0}}{M_{S}}\times 100$$
where *M_S_* is the mass of the swollen gel at equilibrium and *M_0_* is the mass of the dried gel at time *t* = 0.

#### 3.3.2. Network Studies

The cross-link density, *q*, is defined as being the mole fraction of the cross-linked units and is calculated as follows [61,62,63]:(5)$$q=\frac{M_{0}}{M_{C}}$$

*M*_0_ is the molecular weight of the repeating units from a polymer and is calculated using the following relation [61,62,63].
(6)$$M_{0}=\frac{\left(m_{AMD}\times M_{AMD}\right)+\left(m_{AA}\times M_{AA}\right)+(m_{Na-Alg}\times M_{Na-Alg})+\left(m_{TMPT}\times M_{TMPT}\right)}{m_{AMD}+m_{AA}+m_{Na-Alg}+m_{TMPT}}$$
where *m_AMD_*, *m_AA_* and *m_TMPT_* are the masses of *AMD*, *AA*, *Na-Alg*, respectively, with cross-linker expressed in grams and *M_AMD_*, *M_AA_*, *M_Na-Alg_* and *M_TMPT_* are the molar masses of *AMD*, *AA*, *Na-Alg*, respectively, with cross-linker expressed in g mol^−1^ [19].

*M_c_* is the average molar mass between cross-links. According to the theory of Flory and Rehner for a perfect network, *M_c_* is calculated using the following relation [59]:(7)$$M_{C}=-V_{1}d_{P}\frac{\nu_{S}^{1/3}-\nu_{S}/2}{\ln(1-\nu_{S})+\nu_{S}+\chi \nu_{S}^{2}}$$
where *V_1_* is the molar volume of the solvent (water: 18 cm^3^ mol^−1^), *d_P_* is the polymer density, *υ_S_* is the volume fraction of the polymer in the swollen gel (cm^3^) and is equal to 1/*S*, *χ* is the Flory–Huggins interaction parameter between the solvent and polymer. The density of samples (values between 1.045–1.285) was determined by using an electronic balance equipped with kits for density determination (AS 220, 0.1 mg resolution, producer Radwang USA LLC, Radom, Poland) [19]. The value of *χ* is calculated as follows [61,62,63]:(8)$$\chi =0.431-0.311\nu_{S}-0.036\nu_{S}^{2}$$

Other important parameters used for the assessment of networks are the mesh size (*ξ*) and porosity (*P*%). Using the calculated values of *Mc*, the mesh size was determined with the following equation [27]:(9)$$\xi =\upsilon_{s}^{-1/3}l\sqrt{\frac{2C_{n}M_{c}}{M_{r}}}$$
where *υ_S_* is the volume fraction of the polymer in the swollen gel, l is the length of the C-C bond along the polymer backbone (0.154 nm), *C_n_* is the Flory characteristic ratio of the polymer and *M_r_* is the molecular mass of the repeated unit. The characteristic ratio *C_n_*, for Sodium Alginate-g-acrylamide/acrylic Acid hydrogels was taken as the weighted average of C_n_ values for poly(*AMD*) and poly(*AA*) chains. The characteristic ratio *C_n_*, for Sodium Alginate-g-acrylamide/acrylic Acid hydrogels was taken as the weighted average of *C_n_* values for poly(*AMD*), poly(*AA*) and *Na-Alg* chains, according to their molar ratio in the hydrogel (*C_n_* was taken 8.8, 6.7 and 21.1 for poly(*AMD*), poly(*AA*) and *Na-Alg*, respectively) [19,64].

The porosity *P*(%) of the hydrogels was determined using the following equation [59]:(10)$$P(\%)=\frac{V_{d}}{1-V_{d}}\times 100$$
where *V_d_* is the volume ratio of water at equilibrium [19].

#### 3.3.3. Swelling Kinetics and Swelling Power

The controlling mechanism of the swelling processes is usually appreciated through several kinetic models applied to the experimental data. The simplest kinetic analysis is the following first order equation [59]:(11)$$\frac{dS}{dt}=k_{1,S}\left(S_{\max.}-S\right)$$
where *k*_1,*S*_ is the rate constant of first order swelling and *S_max._* is the degree of swelling at equilibrium.

After the definite integration by applying the initial condition (*S* = 0 at *t* = 0 and *S* = *S* at *t* = *t*), the equation becomes:(12)$$\ln W=k_{1,S}t,\;\mathrm{were}\;W=\frac{S\max.}{S_{\max.}-S}$$

For the development of some applications (biomedicine, pharmaceuticals, environmental and agricultural engineering) it is important to clarify the polymer behavior; this can be done by the analysis of water diffusion mechanisms. Water migrates into preexist-ing or dynamically formed spaces between the chains of the hydrogel and this swells. By swelling experiments, the diffusion of water into hydrogels can be determined. By applying the following equations to 60% of swelling curves, the nature of the diffusion of water into hydrogels can be evaluated [65,66].
(13)$$F_{swp}=\frac{M_{t}-M_{0}}{M_{0}}=kt^{n}$$
(14)$$\ln F_{swp}=n\ln t+\ln k$$
where *M_t_* and *M_o_* are the masses of the swollen sample at time *t* and of the dry sample, respectively, *k* is the swelling constant and *n* is the swelling exponent which is an indicator of the transport mechanism.

The swelling–time curves of hydrogels in water are used to calculate the diffusion coefficients (*D*) by the short time approximation method. This method is valid only for the first 60% of the swelling [59]. The diffusion coefficients have been calculated using the following relation:(15)$$F=4{\left[\frac{D}{\pi \times r^{2}}\right]}^{1/2}t^{1/2}$$
where *D* is in cm^2^s^−1^, *t* in s and *r* is the radius of cylindrical polymer sample (cm).

#### 3.3.4. Evaluation of Hydrogels Capacity to Absorb and Release Aqueous Nutrients Solutions by Multiple Usage Experiments

The posibility of multiple use of Sodium Alginate-g-acrylamide/acrylic Acid hydrogels was investigated by successive absorption–release experiments made at room temperature of 25 °C in two different aqueous solutions with nutrients. The hydrogels were immersed in the solutions until they reached constant mass, this being determined by daily weighing. After reaching constant mass, the hydrogels were extracted from the solutions and left to dry until they reached constant mass at a room temperature of 25 °C. After drying, they were re-introduced into the aqueous solutions with nutrients, the previously described procedure being repeated four times. The swelling capacity at equilibrium and the physical stability (mass loss) of the hydrogels were monitored.

#### 3.3.5. Spectral Characterization by Fourier Transform Infrared Spectroscopy (FTIR)

The Sodium Alginate-g-acrylamide/acrylic Acid hydrogels structure was investigated by the Fourier Transform Infrared Spectroscopy (FTIR) technique using the Spectrum 100 instrument (Perkin Elmer, Waltham, MA, USA). The FTIR spectra were acquired in ATR mode in the range of 4000–600 cm^−1^ with a 50 scans/sample and a resolution of 4 cm^−1^. The recorded spectra were analyzed with Spectrum v. 6.3.2 software [67].

#### 3.3.6. Morphological Investigations by Scanning Electron Microscopy (SEM)

The hydrogels were examined by the Scanning Electron Microscopy (SEM) technique using the FEI/Phillips scanning electron microscope (Hillsboro, OR, USA). Samples, fractured in liquid nitrogen and sputtered with gold palladium, were placed in aluminum mounts and scanned at an accelerating voltage of 30 kV [19].

## 4. Conclusions

Sodium Alginate-g-acrylamide/acrylic Acid hydrogels, a possible viable soil conditioner due to the biodegradable potential and capacity of multiple usage, were obtained by electron beam irradiation using the linear accelerator ALID 7 at 5.5 MeV. For this, monomeric solutions with 1 and 2% sodium alginate, acrylamide and acrylic acid in ratios of 1:1 (R2) and 1.5:1 (R1), respectively, were irradiated with 5 and 6 kGy. Specific analyses for the hydrogels physical, chemical, structural and morphological characterization were carried out. Additionally, four successive absorption–release experiments in two different aqueous solutions with nutrients (a synthetic one having an acidic pH and a natural one having a basic pH) were performed. The irradiation dose of 6 kGy has lead to the obtaining of hydrogels with gel fractions of 92.21% and swellings of over 52,800% for hydrogels with 2% *Na-Alg* (H1R2-D2). The equilibrium water content (*EWC*%) and porosity (*P*%) were over 99% for all samples, the closest result to 100% being obtained also for the hydrogels containing 2% *Na-Alg* at the irradiation dose of 6 kGy (EWC of 99.81% and *P* of 99.80% for H2R1-D2). According to the relative rates of diffusion (*R_diff_*) and relaxation (*R_relax_*), by applying the first order swelling equation, a non-Fickian diffusion mechanism was found as being compatible with the hydrogels behavior in water (0.881 < *n* < 0.905). The osmotic pressure of the gel that must be permanently overcome by water during the swelling process may be responsible for the non-Fickian behavior. The posibility of multiple usage of the obtained hydrogels was investigated by successive absorption–release experiments done in two different aqueous solutions with nutrients in terms of of nature and pH: Solution no. 1, a synthetic one, was acidic with a pH of 5.4, while the Solution no. 2, a natural one, was basic with a pH of 7.45. After four absorption–release cycles, swellings of 15,795% and 10,329% were obtained with Solution no. 2 using the hydrogels with the biggest and the smallest gel fractions, respectively (H1R2-D2–92.21% and H2R1-D1–79.55%). In Solution no. 1, using the same hydrogels, swellings were of 9,158% and 8,878%, respectively. FTIR analysis together with reaction mechanisms have confirmed the grafting of acrylamide and acrylic acid on the sodium alginate structure. SEM images show increased sizes of crack-like structures of hydrogels that provide space for solution penetration, a fact that correlated with the high degree of absorption in the water and aqueous solutions with nutrients.

## Figures and Tables

**Figure 1 ijms-24-00104-f001:**
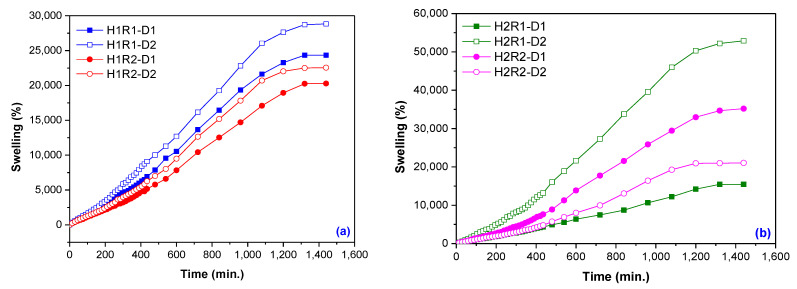
The effect of *Na-Alg* content and irradiation dose on hydrogels swelling: (**a**) 1% *Na-Alg* (H1R1 and H1R2 types); (**b**) 2% *Na-Alg* (H2R1 and H2R2 types).

**Figure 2 ijms-24-00104-f002:**
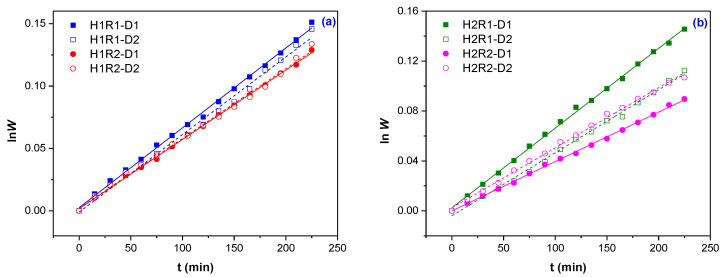
The first order swelling kinetics for the obtained hydrogels: (**a**) 1% *Na-Alg* (H1R1 and H1R2 types); (**b**) 2% *Na-Alg* (H2R1 and H2R2 types).

**Figure 3 ijms-24-00104-f003:**
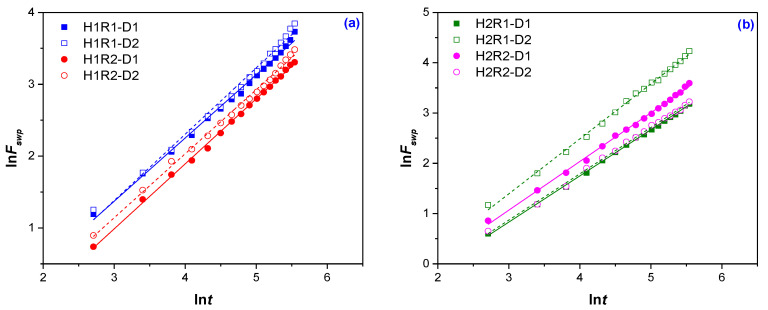
The plots of ln*F* vs. ln*t* for diffusion exponent (*n*) and diffusion constant (*k*) calculation: (**a**) 1% *Na-Alg* (H1R1 and H1R2 types); (**b**) 2% *Na-Alg* (H2R1 and H2R2 types).

**Figure 4 ijms-24-00104-f004:**
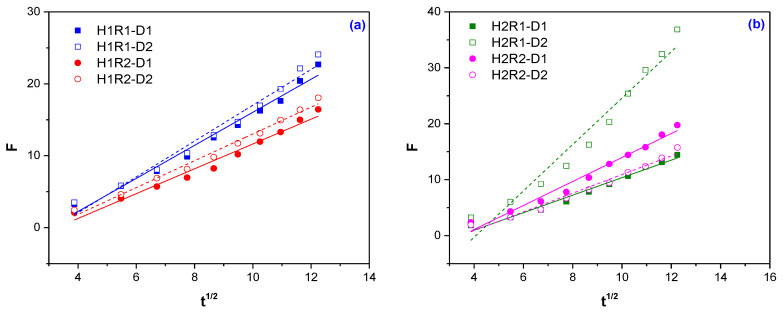
Plots of *F* vs. *t*^1/2^ for diffusional coefficient (*D*) calculation: (**a**) 1% *Na-Alg* (H1R1 and H1R2 types); (**b**) 2% *Na-Alg* (H2R1 and H2R2 types).

**Figure 5 ijms-24-00104-f005:**
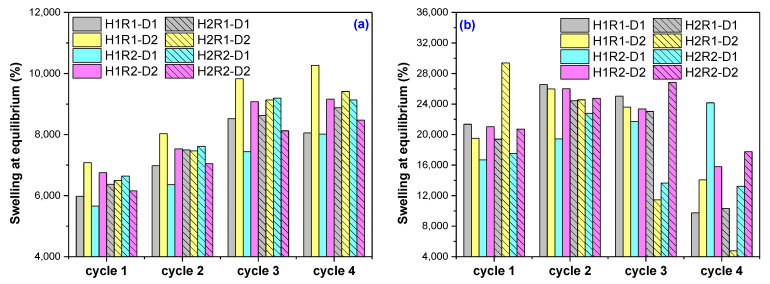
The hydrogels swelling at equilibrium in (**a**) Solution no. 1 (**b**) Solution no. 2.

**Figure 6 ijms-24-00104-f006:**
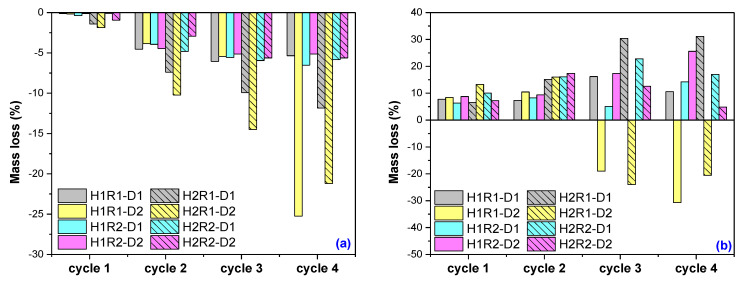
The hydrogels mass loss in (**a**) Solution no. 1 (**b**) Solution no. 2.

**Figure 7 ijms-24-00104-f007:**
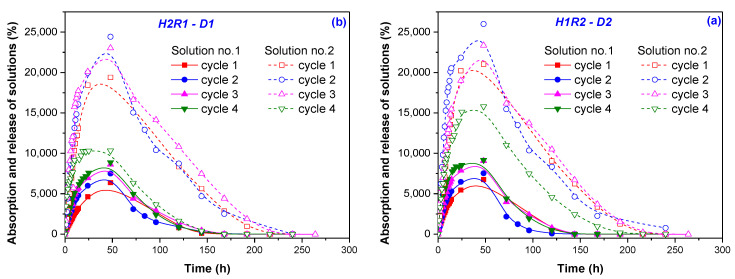
Absorption and release experiments on hydrogels having the (**a**) smallest gel fraction percent (H2R1-D1, 79.55%) and (**b**) biggest gel fraction percent (H1R2-D2, 92.21%).

**Figure 8 ijms-24-00104-f008:**
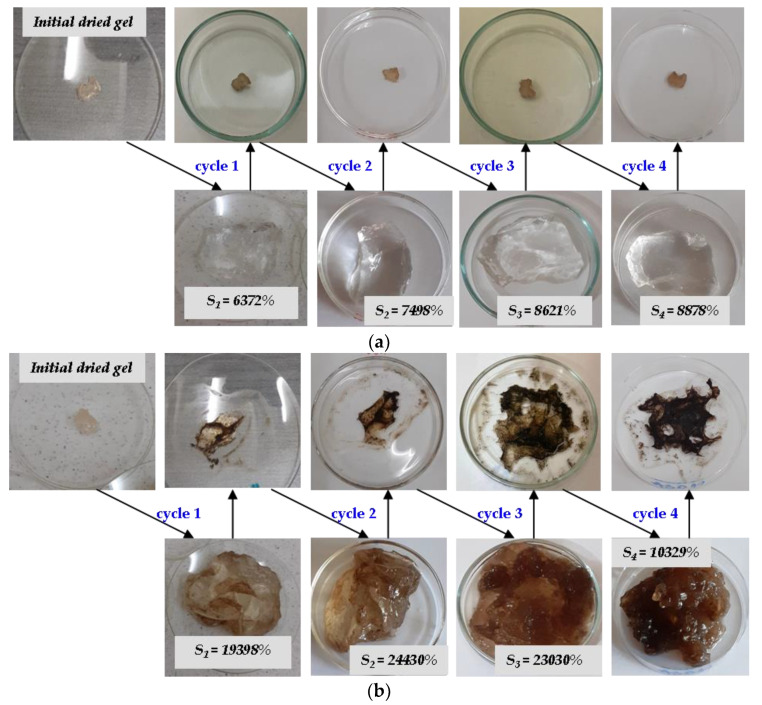
Photographs during the successive absorption–release experiments using the H2R1-D1 hydrogel in (**a**) synthetic Solution no. 1 and (**b**) natural Solution no. 2.

**Figure 9 ijms-24-00104-f009:**
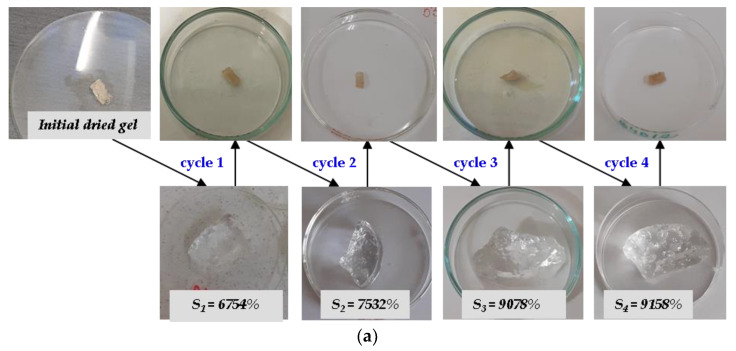
Photographs during the successive absorption–release experiments using the H1R2-D2 hydrogel in (**a**) synthetic Solution no. 1 and (**b**) natural Solution no. 2.

**Figure 10 ijms-24-00104-f010:**
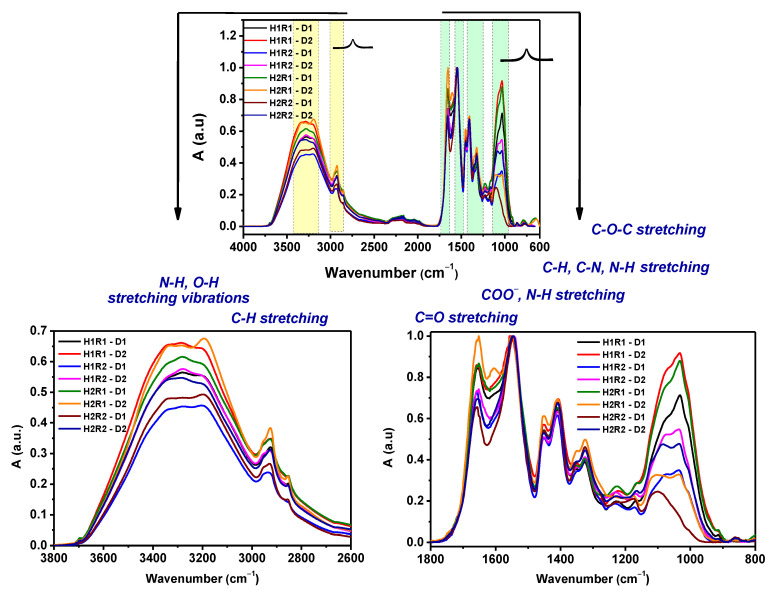
FTIR spectra of dried hydrogels.

**Figure 11 ijms-24-00104-f011:**
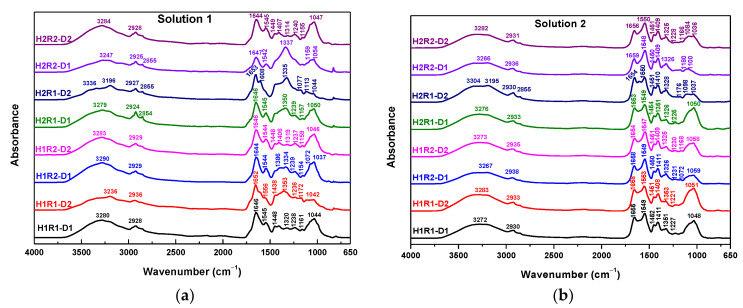
FTIR spectra of the dried hydrogels after the swelling process in (**a**) Solution no. 1 and (**b**) Solution no. 2.

**Figure 12 ijms-24-00104-f012:**
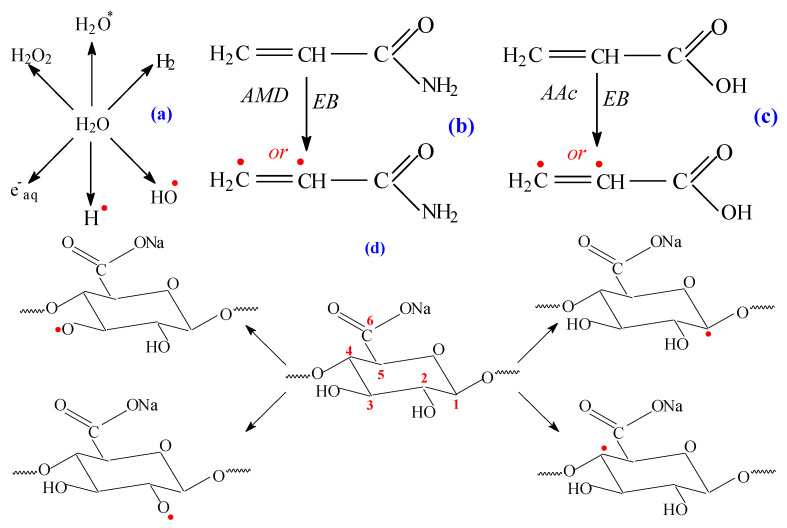
Radiation-induced decomposition of (**a**) solvent (water), (**b**) acrylamide (*AMD*), (**c**) acrylic acid (*AA*) and (**d**) sodium alginate (*Na-Alg*).

**Figure 13 ijms-24-00104-f013:**
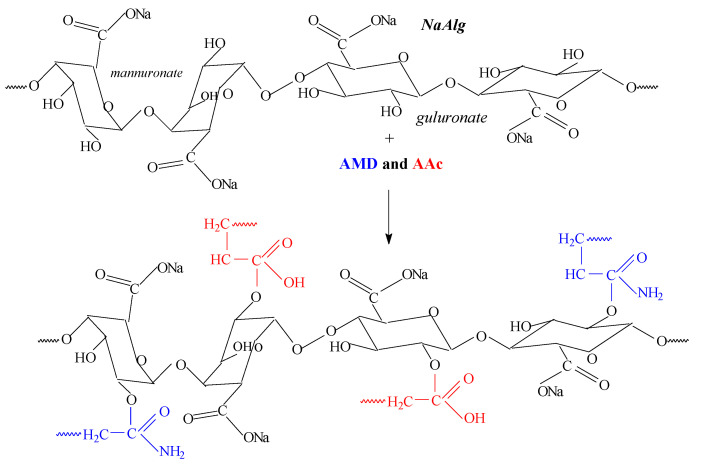
Possible reaction mechanism of acrylamide (*AMD*) and acrylic acid (*AA*) grafting to the sodium alginate (*Na-Alg*) structure.

**Figure 14 ijms-24-00104-f014:**
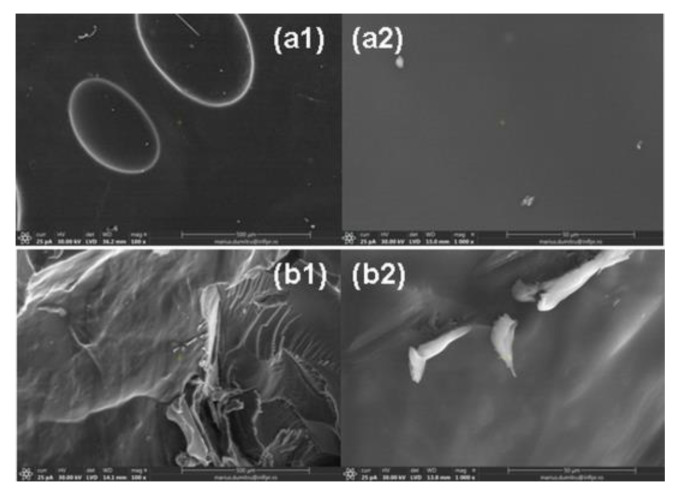
SEM images of (**a1**,**a2**) H2R1-D1 and (**b1**,**b2**) H1R2-D2. Note: (**a1**,**b1**) images captured at the magnification of 100; (**a2**,**b2**) images captured at the magnification of 1000.

**Figure 15 ijms-24-00104-f015:**
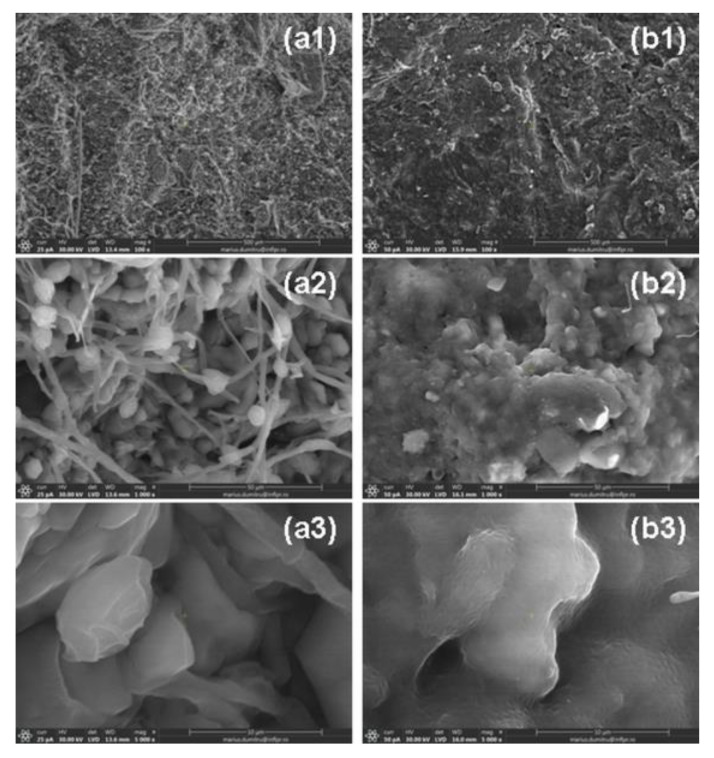
SEM images of H2R1-D1 samples dried after the immersion in (**a1**–**a3**) Solution no. 1 and (**b1**–**b3**) Solution no. 2 at the magnifications of (**a1**,**b1**) 100, (**a2**,**b2**) 1000 and (**a3**,**b3**) 5000.

**Figure 16 ijms-24-00104-f016:**
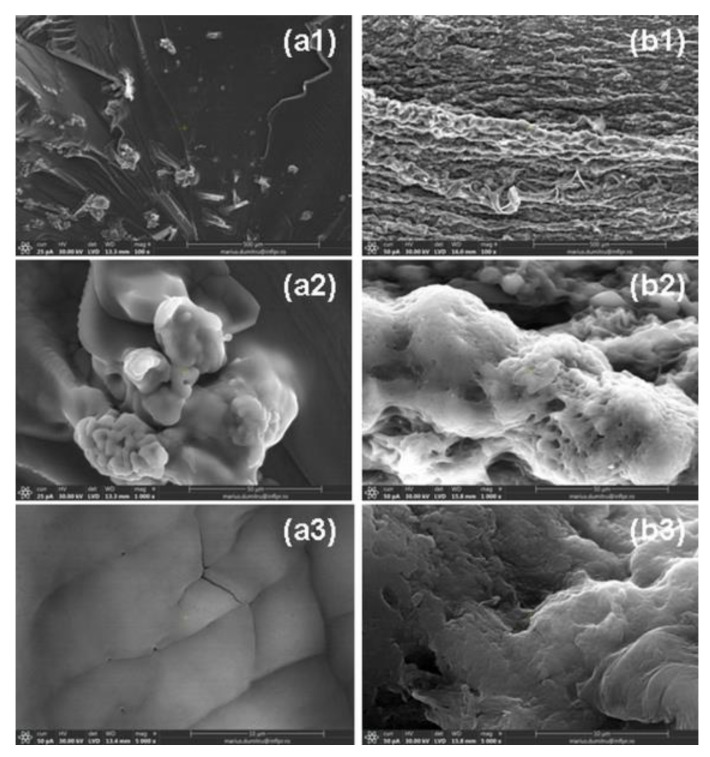
SEM images of H1R2-D2 samples dried after the immersion in (**a1**–**a3**) Solution no. 1 and (**b1**–**b3**) Solution no. 2. Note: (**a1**,**b1**) images captured at the magnification of 100; (**a2**,**b2**) images captured at the magnification of 1000, (**a3**,**b3**) images captured at the magnification of 5000.

**Table 1 ijms-24-00104-t001:** Gel fraction, swelling and equilibrium water content measurements results.

Samples Codes	Gel Fraction (%)	Swelling (%)	EWC (%)
H1R1-D1	81.39	24,347	99.59
H1R2-D1	89.46	20,276	99.51
H2R1-D1	79.55	15,460	99.36
H2R2-D1	80.76	35,194	99.72
H1R1-D2	82.64	28,826	99.65
H1R2-D2	92.21	22,555	99.56
H2R1-D2	81.75	52,882	99.81
H2R2-D2	87.35	21,029	99.53

**Table 2 ijms-24-00104-t002:** Cross-link density, porosity and mesh size measurements results.

Samples Codes	Cross-Link Density (*q* × 10^5^)	Porosity (%)	Mesh Size, ξ (nm)
H1R1-D1	1.09	99.74	486
H1R2-D1	1.66	99.65	371
H2R1-D1	1.42	99.73	432
H2R2-D1	5.30	99.79	844
H1R1-D2	0.77	99.77	611
H1R2-D2	1.50	99.64	396
H2R1-D2	0.72	99.80	688
H2R2-D2	1.42	99.67	431

**Table 3 ijms-24-00104-t003:** The values of swelling rate constants (*k*_1,S_/min^−1^), diffusion exponent (*n*), diffusion constant (*k*) and diffusional coefficient (*D*/cm^2^s^−1^) obtained after the first order swelling approximation.

Samples Codes	First Order Swelling Rate Constants		Diffusion				
			Mechanisms of Water Diffusion			Diffusion Coefficient	
	*k*_1,*S*_ × 10^3^	*R* ^2^	*n*	*k*	*R^2^*	*D* × 10^3^	*R* ^2^
H1R1-D1	0.640	0.998	0.881	0.281	0.995	4.321	0.979
H1R1-D2	0.622	0.994	0.923	0.250	0.990	5.070	0.974
H1R2-D1	0.555	0.999	0.905	0.178	0.998	2.435	0.975
H1R2-D2	0.559	0.995	0.891	0.216	0.996	2.843	0.987
H2R1-D1	0.639	0.998	0.919	0.146	0.998	0.501	0.981
H2R1-D2	0.505	0.996	1.093	0.152	0.997	3.517	0.953
H2R2-D1	0.396	0.998	0.968	0.159	0.997	0.939	0.976
H2R2-D2	0.482	0.997	0.921	0.151	0.998	0.571	0.976

**Table 4 ijms-24-00104-t004:** Attribution of main bands in FTIR spectra of *Na-Alg*, *AMD* and hydrogel.

Wavenumber cm^−1^			Band Assignments
*Na-Alg*	*AMD*	Hydrogel	
–	3360	3330–3345	-H stretching vibration
3280	3190	3190–3200	-OH stretching vibration
2950	2930/2860	2930–2940	C-H, CH_2_ stretching vibration
–	1654	1649–1655	C=O stretching vibration
1620	1613	1600–1620	-COO^-^ symmetric bending
1417	1429	1415–1450	-COO^-^ symmetric bending
–	1353	–	-CN stretching vibration
1349	–	1345–1350	CH_2_, C-H bending mode
1120	1121	1105–1120	C-O-C stretching
1030	1037	1032–1045	C-O stretching vibrations

**Table 5 ijms-24-00104-t005:** Physical and chemical properties of materials used for Sodium Alginate-g-acrylamide/acrylic Acid hydrogels preparation.

Row Materials	Chemical Characteristics
Sodium alginate, *Na-Alg* (C_6_H_7_NaO_6_)_n_ or C_6_H_9_NaO_7_	- molecular weight: 216.121 g/mol; - density: 1.601 g/cm^3^; - solubility in water: no more than 2% on the dried basis;
Acrylic acid, *AA* C_3_H_4_O_2_	- molecular weight: 71.08 g/mol; - density: 1.13 g/cm^3^; - soluble in water;
Acrylamide, *AMD* C_3_H_5_NO	- molecular weight: 72.06 g/mol; - density: 1.051 g/cm^3^; - solubility in water: 2.04 kg L^−1^ at 25 °C;
Potassium persulfate, *PP* (used as reaction initiator) K_2_S_2_O_8_	- molecular weight: 270.322 g/mol; - density: 2.477 g/cm^3^; - solubility in water: 1.75 g/100 mL at 0 °C;
Trimethylolpropane trimethacrylate, *TMPT* (used as cross-linker) C_18_H_26_O_6_	- molecular weight: 338.4 g/mol; - boiling point: >200 °C; - density 1.06 g/cm^3^; - 75 ± 3% active ingredient;

**Table 6 ijms-24-00104-t006:** The aqueous nutrients solutions and their characteristics used for Sodium Alginate-g-acrylamide/acrylic Acid hydrogels testing in succesive absorption and release experiments.

Nutrients Solutions	Characteristics
Solution no. 1: Liquid fertilizer for balcony flowers (produced by AGRO CS, Lucenec, Slovakia); used according to the manufacturer instructions: 15 mL diluted in 1000 mL water	- synthetic product; - pH = 5.4; - nitrogen 7%; - phosphorus 4%; - potassium 5%;
Solution no. 2: Biopon natural biohumus for vegetables and greens (produced by Bros Sp. z o.o. sp. k., Poznan, Poland); used according to the manufacturer instructions: 120 mL diluted in 1000 mL water	- organic product, 100% natural; - pH = 7.45. - made from vermicompost, it is the final product of the decomposition of organic matter by earthworms, which during the digestion of the soil, saturates it with beneficial microorganisms, thus releasing nutrients from the soil.

**Table 7 ijms-24-00104-t007:** Sodium Alginate-g-acrylamide/acrylic Acid hydrogels synthesis details.

Samples Codes	Amount of Chemicals (g/100 mL Solution)					Irradiation Dose (kGy)
	*Na-Alg*	*AA*	*AMD*	*PP*	*TMPT*	
H1R1-D1	1	12.5	18.75	0.025	0.02	5
H1R1-D2						6
H1R2-D1		18.75	18.75	0.025	0.02	5
H1R2-D2						6
H2R1-D1	2	12.5	18.75	0.025	0.02	5
H2R1-D2						6
H2R2-D1		18.75	18.75	0.025	0.02	5
H2R2-D2						6

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
