# Peer review of "Sodium Alginate-g-acrylamide/acrylic Acid Hydrogels Obtained by Electron Beam Irradiation for Soil Conditioning"

_ijms, 2022, doi:10.3390/ijms24010104_

Round 1
Reviewer 1 Report
The authors are capable of studying ionizing radiation polymerization hydrogels, however, to be published in a high standard journal like IJMS, the characterization and application of the materials are still relatively crude, especially for SA,PAM,PAA, which have been extensively studied, and it is recommended that the authors collaborate with a materials science department to enhance their study. Specific suggestions are as follows.
1. I think scanning electron microscopy data should be added to confirm the internal network structure changes of hydrogel. Solid characterizations will be more comprehensive to reveal the hydrogel water absorption performance.
2. It is recommended that the authors use the IJMS template for better review and editing.
3. Some figures/tables are suggested to be placed in the Supporting Materials.
4. Some discussions are tendentious and the authors should use objective descriptions.
5. Manuscript should be polished.
I would like to review the revised paper.
Author Response
Accordingly to the reviewers and editor requests the changes that have been made in the manuscript are marked in track changes mode.
Thus,
- English style has been improved in accordance with the suggestions that have been done by both Reviewers;
- In the Introduction a series of paragraphs have been rewritten in accordance with the suggestions that have been done by both Reviewers. We would also show the difference between chemical polymerization (through covalent bonds obtained by electron beam irradiation in this paper) and physical polymerization (by multi-valent ions) as well as the role of the synthetic monomers and biopolymer in the hydrogel structure, in accordance with the suggestions and comments that have been done by Reviewer 2.
- In the subchapter 3.1 Percentages of Gel Fraction, Swelling and Equilibrium Water Content the Table 4 was modified in accordance with the suggestions that have been done by Reviewer 1. We also explained in what way a slight increase in the radiation dose (from 5 to 6 kGy) can induces a strong (by a factor of 3.5) increase in the equilibrium degree of swelling without the extra extra cross-links effect appearing, in accordance with the suggestions and comments that have been done by Reviewer 2. Some paragraph has been rewritten in accordance with the suggestions that have been done by both Reviewers;
- In the subchapter 3.2. Cross-link density, porosity and mesh size the Table 5 was modified in accordance with the suggestions that have been done by Reviewer 1. Some paragraph has been rewritten in accordance with the suggestions that have been done by both Reviewers;;
- In the subchapter 3.3. Swelling Kinetcs, Swelling Power and Diffusion coefficients Determination some paragraph has been rewritten in accordance with the suggestions that have been done by both Reviewers. We also explained how that the diffusion exponent “n” obtained is close to unity for all gels under investigation in accordance with the suggestions and comments that have been done by Reviewer 2.
- In the subchapter 3.4. Absorption and Release Capacities some paragraph has been rewritten in accordance with the suggestions that have been done by both Reviewers.
- A new subchapter named 3.6. Morphological characterization was introduced in accordance with the suggestions and comments that have been done by Reviewer 1.
- The manuscript has been rewritten with the suggestions that have been done by both Reviewers;
- A number of references have been replaced.
- The manuscript was written in the ijms format

Reviewer 2 Report
The study deals with preparation and the experimental investigation of the physical properties of hydrogels based on alginate, acrylamide and acrylic acid chains bridged by covalent cross-links and subjected to electron beam irradiation. This work extends the previous study [40], where a similar analysis was performed on gels not containing sodium alginate chains.
Although preparation of superabsorbent bio-based hydrogels for applications in agriculture is an important topic, I cannot recommend this paper for publication in the present form for the following reasons:
1. The hydrogels prepared by the authors are synthetic (not bio-based), with a small addition of sodium alginate only (the concentration of sodium alginate in the network was below 6.5 wt.% for all conditions of synthesis). A reason for this addition and the role of sodium alginate in the swelling process are not explained. Specifically, it remains unclear why alginate chains are not cross-linked by multi-valent ions under preparation (the standard procedure).
2. The authors report that a weak increase in the radiation dose (from 5 to 6 kGy) induces a strong (by a factor of 3.5) increase in the equilibrium degree of swelling (Samples H2R1-D1 and H2R1-D2 in Tab. 4). This result contradicts their own statement that EB irradiation causes formation of cross-links between chains, because the presence of extra cross-links leads to a decrease (not increase) in degree of swelling.
3. The authors claim that the diffusion exponent “n” is close to unity for all gels under investigation (Tab. 6). This result contradicts majority of previous studies where the conventional Fickian diffusion of water was observed under swelling. An explanation is required for this finding.
Author Response

(The authors gave the same response as above.)

Round 2
Reviewer 1 Report
Publish as is.
Reviewer 2 Report
The authors improved substantially the exposition. I am pleased to recommend publication of this work.